# Multi-Time Attention Networks for Irregularly Sampled Time Series

**Satya Narayan Shukla** [1]   **Benjamin M. Marlin** [1]

## Abstract

Irregular sampling occurs in many time series modeling applications where it presents a significant challenge to standard deep learning models. This work is motivated by the analysis of physiological time series data in electronic health records, which are multivariate, sparse, irregularly sampled, and incompletely observed. In this paper, we propose a new deep learning framework for this setting that we call Multi-Time Attention Networks, which use embeddings and attention to produce fixed-dimensional representations of irregularly sampled multivariate time series. We evaluate this framework through applications to both interpolation and classification and show that it outperforms several recently proposed methods while offering significantly faster training times than current state-of-the-art approaches.

## 1. Introduction

In this paper we consider the problem of modeling time series data that are multivariate, sparse, and irregularly sampled. Irregular sampling generalizes missingness in sequential data to the case of univariate continuous-time time series data. In the case of multivariate irregularly sampled time series, it is also commonly the case that only a subset of data dimensions are observed at any given time point. Such data occur in applications including healthcare, climate science, ecology, astronomy, biology and others.

It is well understood that irregular sampling poses a significant challenge to machine learning models, which typically assume fully-observed, fixed-size feature representations (Yadav et al., 2018). While recurrent neural networks (RNNs) have been widely used to model such data because of their ability to handle variable length sequences, basic RNNs do not account for irregular spacing between observation times or a lack of alignment of the time points where observations for different variables occur. However, both of these problems can hold for real-world sparse and irregularly observed time series. To respond to these challenges, there has been significant progress over the last decade on building and adapting machine learning models that can better capture the structure of irregularly sampled multivariate time series (Marlin et al., 2012; Li & Marlin, 2015; 2016; Lipton et al., 2016; Futoma et al., 2017; Che et al., 2018; Shukla & Marlin, 2019; Rubanova et al., 2019).

In this work, we introduce a new model for multivariate, sparse and irregularly sampled time series that we refer to as *Multi-Time Attention networks* or mTANs. mTANs are fundamentally continuous-time interpolation-based models. Their primary innovations are the inclusion of a learned continuous-time embedding mechanism coupled with a time attention mechanism that replaces the use of a fixed similarity kernel when forming representation from continuous time inputs. This gives mTANs more representational flexibility than previous interpolation-based models.

Our approach re-represents an irregularly sampled time series at a fixed set of reference points. The proposed time attention mechanism uses reference time points as queries and the observed time points as keys. We propose an encoder-decoder framework for end-to-end learning using an mTAN module to interface with given multivariate, sparse and irregularly sampled time series inputs. The encoder takes the irregularly sampled time series as input and produces a fixed-length latent representation over a set of reference points, while the decoder uses the latent representations to produce reconstructions conditioned on the set of observed time points. Learning uses methods established for variational autoencoders (Kingma & Welling, 2014). We evaluate the performance of the proposed framework on interpolation and classification tasks. We compare the proposed framework to several deep learning baselines built on gated recurrent unit (GRU) models using simple interpolation/imputation approaches. We also compare to current state-of-the-art interpolation and classification methods designed specifically for irregularly sampled time series. Our approach performs better than a range of baseline and recently proposed models while offering significantly faster training times than current state-of-the-art methods.

[1]College of Information and Computer Sciences, University of Massachusetts Amherst, Amherst, MA, USA. Correspondence to: Satya Narayan Shukla <snshukla@cs.umass.edu>.

*Presented at the first Workshop on the Art of Learning with Missing Values (Artemiss) hosted by the $37^{th}$ International Conference on Machine Learning (ICML).* Copyright 2020 by the author(s).

## 2. The Multi-Time Attention Module

In this section, we present the proposed Multi-Time Attention Module (mTAN). The role of this module is to re-represent a sparse and irregularly sampled time series in a fixed-dimensional space. This module uses multiple continuous-time embeddings and attention-based interpolation. We begin by presenting notation followed by the time embedding and attention components.

**Notation:** In the case of a supervised learning task, we let $\mathcal{D} = \{(\mathbf{s}_n, y_n)|n = 1, ..., N\}$ represent a data set containing $N$ data cases. An individual data case consists of a single target value $y_n$ (discrete for classification), as well as a $D$-dimensional, sparse and irregularly sampled multivariate time series $\mathbf{s}_n$. Different dimensions $d$ of the multivariate time series can have observations at different times, as well as different total numbers of observations $L_{dn}$. Thus, we represent time series $d$ for data case $n$ as a tuple $\mathbf{s}_{dn} = (\mathbf{t}_{dn}, \mathbf{x}_{dn})$ where $\mathbf{t}_{dn} = [t_{1dn}, ..., t_{L_{dn}dn}]$ is the list of time points at which observations are defined and $\mathbf{x}_{dn} = [x_{1dn}, ..., x_{L_{dn}dn}]$ is the corresponding list of observed values. In the case of an unsupervised task such as interpolation, each data case consists of a multivariate time series $\mathbf{s}_n$ only. We drop the data case index $n$ for brevity when the context is clear.

**Time Embedding:** Time attention module is based on embedding continuous time points into a vector space. We generalize the notion of a positional encoding used in transformer-based models to continuous time. Time attention networks simultaneously leverage $H$ embedding functions $\phi_h(t)$, each outputting a representation of size $d_r$. Dimension $i$ of embedding $h$ is defined as follows:

$$\phi_h(t)[i] = \begin{cases} \omega_{0h} \cdot t + \alpha_{0h}, & \text{if} \quad i = 0 \\ \sin(\omega_{ih} \cdot t + \alpha_{ih}), & \text{if} \quad 0 < i < d_r \end{cases} \quad (1)$$

where the $\omega_{ih}$'s and $\alpha_{ih}$'s are learnable parameters. The periodic terms capture the periodicity in the time series. In this case, $\omega_{ih}$ and $\alpha_{ih}$ represent the frequency and phase of the sine function. The linear term, on the other hand, can capture non-periodic patterns dependent on the progression of time. For a given difference $\Delta$, $\phi_h(t + \Delta)$ can be represented as a linear function of $\phi_h(t)$.

Learning the periodic time embedding functions is equivalent to using a one-layer fully connected network with a sine function non-linearity to map the time values into a higher dimensional space. By contrast, positional encoding used in transformer models is defined only for discrete positions. We note that our time embedding functions can subsume positional encodings when evaluated at discrete positions.

**Multi-Time Attention:** The time embedding component described above takes a continuous time point and embeds

it into $H$ different $d_r$-dimensional spaces. In this section, we describe how we leverage time embeddings to produce a continuous-time embedding module for sparse and irregularly sampled time series. This *multi-time attention* embedding module mTAN$(t, \mathbf{s})$ takes as input a query time point $t$ and a set of keys and values in form of $D$-dimensional multivariate sparse and irregularly sampled time series $\mathbf{s}$ (as defined in the notation section above), and returns a $J$ dimensional embedding at time $t$. This process leverages a continuous-time attention mechanism applied to the $H$ time embeddings. The complete computation is described below.

$$\text{mTAN}(t, \mathbf{s})[j] = \sum_{h=1}^{H} \sum_{d=1}^{D} \hat{x}_{hd}(t, \mathbf{s}) \cdot U_{hdj} \quad (2)$$

$$\hat{x}_{hd}(t, \mathbf{s}) = \sum_{i=1}^{L_d} \kappa_h(t, t_{id}) \, x_{id} \quad (3)$$

$$\kappa_h(t, t_{id}) = \frac{\exp\left(\phi_h(t)WV^T\phi_h(t_{id})^T/\sqrt{d_k}\right)}{\sum_{i'=1}^{L_d} \exp\left(\phi_h(t)wv^T\phi_h(t_{i'd})^T/\sqrt{d_k}\right)} \quad (4)$$

As shown in Equation 2, dimension $j$ of the mTAN embedding mTAN$(t, \mathbf{s})[j]$ is given by a linear combination of intermediate univariate continuous-time functions $\hat{x}_{hd}(t, \mathbf{s})$. There is one such function defined for each input data dimension $d$ and each time embedding $h$. The parameters $U_{hdj}$ are learnable linear combination weights.

As shown in Equation 3, the structure of the intermediate continuous-time function $\hat{x}_{hd}(t, \mathbf{s})$ is essentially a kernel smoother applied to the $d^{th}$ dimension of the time series. However, the interpolation weights $\kappa_h(t, t_{id})$ are defined based on a time attention mechanism that leverages time embeddings, as shown in Equation 4. As we can see, the same time embedding function $\phi_h(t)$ is applied for all data dimensions. The form of the attention mechanism is a softmax function over the observed time points $t_{id}$ for dimension $d$. The activation within the softmax is a scaled inner product between the time embedding $\phi_h(t)$ of the query time point $t$ and the time embedding $\phi_h(t_{id})$ of the observed time point, the key. The parameters $W$ and $V$ are each $d_r \times d_k$ matrices where $d_k \leq d_r$. We use a scaling factor $\frac{1}{\sqrt{d_k}}$ to normalize the dot product to counteract the growth in the dot product magnitude with increase in the dimension $d_k$.

Learning the time embeddings provides our model with flexibility to learn complex temporal kernel functions $\kappa_h(t, t')$. The use of multiple simultaneous time embeddings $\phi_h(t)$ and a final linear combination across time embedding dimensions and data dimensions means that the final output representation function mTAN$(t, \mathbf{s})$ is extremely flexible. Different input dimensions can leverage different time embeddings via learned sparsity patterns in the parameter tensor $U$. Information from different data dimensions can also be mixed together to create compact reduced dimensional representations. We note that all of the required computations can be parallelized using masking variables to deal with unobserved dimensions.

**Discretization:** Since the mTAN module defines a multi-variate function of a continuous time input $t$, $\text{mTAN}(t, \mathbf{s})$, it can not be directly incorporated into neural network architectures that expect inputs in the form of fixed-dimensional vectors or discrete sequences. However, the mTAN module can easily be adapted to produce such an output representation by materializing its output at a set of reference time points $\mathbf{r} = [r_1, ..., r_T]$. In some cases, we may have a fixed set of such points. In other cases, the set of reference time points may need to depend on $\mathbf{s}$ itself. In particular, we define the auxiliary function $\rho(\mathbf{s})$ to return the set of time points where there is an observation on any dimension of $\mathbf{s}$.

Given a collection of reference time points $\mathbf{r}$, we define the discretized mTAN module $\text{mTAND}(\mathbf{r}, \mathbf{s})$ as $\text{mTAND}(\mathbf{s})[i] = \text{mTAN}(r_i, \mathbf{s})$. This module takes as input the set of reference time points $\mathbf{r}$ and the time series $\mathbf{s}$ and outputs a sequence of mTAN embeddings of length $|\mathbf{r}|$, each of dimension $J$. The mTAND module can be used to interface sparse and irregularly sampled multivariate time series data with any deep neural network layer type including fully-connected, recurrent, and convolutional layers. In the next section, we describe the construction of a temporal encoder-decoder architecture leveraging the mTAND module, which can be applied to both classification and interpolation tasks.

## 3. Encoder-Decoder Framework

As described in the last section, we leverage the discretized mTAN module in an encoder-decoder framework as our primary model in this paper, which we refer to as an mTAN network. We develop the encoder-decoder framework within the variational autoencoder (VAE) framework.

**Model Architecture:** As we are modeling time series data, we begin by defining a sequence of latent states $\mathbf{z}_i$. Each of these latent states are iid-distributed according to a standard multivariate normal distribution $p(\mathbf{z}_i)$. We let the set of latent states be $\mathbf{z} = [\mathbf{z}_1, ..., \mathbf{z}_K]$ defined at $K$ reference time points.

We define a three-stage decoder. First, the latent states are processed through an RNN decoder module to induce temporal dependencies, resulting in a first set of deterministic latent variables $\mathbf{h}_{RNN}^{dec} = [\mathbf{h}_{1,RNN}^{dec}, ..., \mathbf{h}_{K,RNN}^{dec}]$. Second, the output of the RNN decoder stage and the $K$ time points $\mathbf{h}_{RNN}^{dec}$ are provided to the mTAND module along with a set of $T$ query time points $\mathbf{t}$. The mTAND module outputs a sequence of embeddings $\mathbf{h}_{TAN}^{dec} = [\mathbf{h}_{1,TAN}^{dec}, ..., \mathbf{h}_{T,TAN}^{dec}]$ of length $|\mathbf{t}|$. Third, the mTAN embeddings are independently decoded using a fully connected decoder $f^{dec}()$ and the result is used to parameterize an output distribution. In this work, we use a diagonal covariance Gaussian distribution with mean given by the final decoded representation.

The final generated time series is given by $\mathbf{s} = (\mathbf{t}, \mathbf{x})$ with all data dimensions observed. The full generative process is shown below. We let $p_\theta(\mathbf{x}|\mathbf{z}, \mathbf{t})$ define the probability distribution over the values of the time series $\mathbf{x}$ given the time points $\mathbf{t}$ and the latent variables $\mathbf{z}$. $\theta$ represents the parameters of all components of the decoder.

$$\mathbf{z}_k \sim p(\mathbf{z}_k) \tag{5}$$

$$\mathbf{h}_{RNN}^{dec} = \text{RNN}^{dec}(\mathbf{z}) \tag{6}$$

$$\mathbf{h}_{TAN}^{dec} = \text{mTAND}^{dec}(\mathbf{t}, \mathbf{h}_{RNN}^{dec}) \tag{7}$$

$$x_{id} \sim p(\mathbf{x}_{id}|f^{dec}(\mathbf{h}_{i,TAN}^{dec})[d]) \tag{8}$$

For an encoder, we simply invert the structure of the generative process. We begin by mapping the input time series $\mathbf{s}$ through the mTAND module along with a collection of $K$ reference time points $\mathbf{r}$. We apply an RNN encoder to the mTAND model that outputs $\mathbf{h}_{TAN}^{enc}$ to encode longer-range temporal structure. Finally, we construct a distribution over latent variables at each reference time point using a diagonal Gaussian distribution with mean and variance output by fully connected layers applied to the RNN outputs $\mathbf{h}_{RNN}^{enc}$. The complete encoder architecture is described below. We define $q_\gamma(\mathbf{z}|\mathbf{r}, \mathbf{s})$ to be the distribution over the latent variables induced by the input time series $\mathbf{s}$ and the reference time points $\mathbf{r}$. $\gamma$ represents all of the parameters in all of the encoder components.

$$\mathbf{h}_{TAN}^{enc} = \text{mTAND}^{enc}(\mathbf{r}, \mathbf{s}) \tag{9}$$

$$\mathbf{h}_{RNN}^{enc} = \text{RNN}^{enc}(\mathbf{h}_{TAN}^{enc}) \tag{10}$$

$$\mathbf{z}_k \sim q(\mathbf{z}_k|\boldsymbol{\mu}_k, \boldsymbol{\sigma}_k^2) \tag{11}$$

$$\boldsymbol{\mu}_k = f_\mu^{enc}(\mathbf{h}_{k,RNN}^{enc}), \ \ \boldsymbol{\sigma}_k^2 = \exp(f_\sigma^{enc}(\mathbf{h}_{k,RNN}^{enc})) \tag{12}$$

**Unsupervised Learning:** To learn the parameters of our encoder-decoder model given a data set of sparse and irregularly sampled time series, we follow a slightly modified VAE training approach and maximize a normalized variational lower bound on the log marginal likelihood based on the evidence lower bound or ELBO. The learning objective is defined below where $p_\theta(x_{jdn}|\mathbf{z}, \mathbf{t}_n)$ and $q_\gamma(\mathbf{z}|\mathbf{r}, \mathbf{s}_n)$ are defined in the previous section.

$$\mathcal{L}_{\text{NVAE}}(\theta, \gamma) = \sum_{n=1}^{N} \frac{1}{\sum_d L_{dn}} \Big(\mathbb{E}_{q_\gamma(\mathbf{z}|\mathbf{r}, \mathbf{s}_n)}[\log p_\theta(\mathbf{x}_n|\mathbf{z}, \mathbf{t}_n)]$$

$$- D_{\text{KL}}(q_\gamma(\mathbf{z}|\mathbf{r}, \mathbf{s}_n)||p(\mathbf{z}))\Big) \tag{13}$$

$$D_{\text{KL}}(q_\gamma(\mathbf{z}|\mathbf{r}, \mathbf{s}_n)||p(\mathbf{z})) = \sum_{i=1}^{T} D_{\text{KL}}(q_\gamma(\mathbf{z}_i|\mathbf{r}, \mathbf{s}_n)||p(\mathbf{z}_i))$$

$$\log p_\theta(\mathbf{x}_n|\mathbf{z}, \mathbf{t}_n) = \sum_{d=1}^{D} \sum_{j=1}^{L_{dn}} \log p_\theta(x_{jdn}|\mathbf{z}, t_{jdn}) \tag{14}$$

Since irregularly sampled time series can have different numbers of observations across different dimensions as well

as across different data cases, it can be helpful to normalize the terms in the standard ELBO objective to avoid the model focusing on sequences that are longer at the expense of shorter sequences. The objective above normalizes the contribution of each data case by the total number of observations it contains. The fact that all data dimensions are not observed at all time points is accounted for in Equation 14. In practice, we use $k$ samples from the variational distribution $q_\gamma(\mathbf{z}|\mathbf{r}, \mathbf{s}_n)$ to compute the learning objective.

**Supervised Learning:** We can also augment the encoder-decoder model with a supervised learning component that leverages the latent states as a feature extractor. We define this component to be of the form $p_\delta(y_n|\mathbf{z})$ where $\delta$ are the model parameters. This leads to an augmented learning objective as shown in equation below where the $\lambda$ term trades off the supervised and unsupervised terms.

$$\mathcal{L}_{\text{sup}}(\theta, \gamma, \delta) = \mathcal{L}_{\text{NVAE}}(\theta, \gamma) + \lambda \mathbb{E}_{q_\gamma(\mathbf{z}|\mathbf{r}, \mathbf{s}_n)} \log p_\delta(y_n|\mathbf{z})$$

In this work, we focus on classification as an illustrative supervised learning problem. For the classification model $p_\delta(y_n|\mathbf{z})$, we use a GRU followed by a 2-layer fully connected network. We use a small number of samples to approximate the required intractable expectations during both learning and prediction. Predictions are computed by marginalizing over the latent variable as shown below.

$$y^* = \arg\max_{y \in \mathcal{Y}} \mathbb{E}_{q_\gamma(\mathbf{z}|\mathbf{r}, \mathbf{s})}[\log p_\delta(y|\mathbf{z})] \qquad (15)$$

## 4. Experiments

In this section, we present interpolation and classification experiments on the Physionet Challenge 2012 dataset. Details about the data set and experimental protocols are described in Appendix A.1 and A.2. We compare the performance of the proposed encoder-decoder model based on the discretized multi-time attention module (**mTAND-Full**) and an ablated model based only on the mTAND encoder (**mTAND-Enc**) to several GRU based interpolation/imputation approaches as well as current state-of-the-art methods (L-ODE-ODE (Rubanova et al., 2019), IP-Nets (Shukla & Marlin, 2019)). Baseline methods are described in detail in Appendix A.3.

Table 1 compares the performance of all methods on the interpolation task defined in Rubanova et al. (2019). As we can see, the proposed method (mTAND-Full) substantially outperforms all of the previous approaches. Table 2 compares predictive performance on the PhysioNet mortality prediction task. The full multi-time attention network model (mTAND-Full) and the classifier based only on the multi-time attention network encoder (mTAND-Enc) achieve significantly improved performance relative to the current state-of-the-art methods (ODE-RNN and L-ODE-ODE) and other baseline methods. We also report the time

*Table 1.* PhysioNet: Interpolation

| Model | MSE $(\times 10^{-3})$ |
|---|---|
| RNN-Impute | $3.243 \pm 0.275$ |
| RNN-$\Delta_t$ | $3.520 \pm 0.276$ |
| RNN-Decay | $3.215 \pm 0.276$ |
| RNN GRU-D | $3.384 \pm 0.274$ |
| RNN-VAE | $5.390 \pm 0.249$ |
| ODE-RNN | $2.361 \pm 0.086$ |
| L-ODE (RNN) | $3.907 \pm 0.252$ |
| L-ODE (ODE) | $2.118 \pm 0.271$ |
| **mTAND-Full** | $\mathbf{0.424 \pm 0.018}$ |

*Table 2.* PhysioNet: Classification

| Model | AUC Score | time |
|---|---|---|
| RNN-Impute | $0.764 \pm 0.016$ | 0.5 |
| RNN-$\Delta_t$ | $0.787 \pm 0.014$ | 0.5 |
| RNN-Decay | $0.807 \pm 0.003$ | 0.7 |
| RNN GRU-D | $0.818 \pm 0.008$ | 0.7 |
| RNN-VAE | $0.515 \pm 0.040$ | 2.0 |
| ODE-RNN | $0.833 \pm 0.009$ | 16.5 |
| L-ODE-RNN | $0.781 \pm 0.018$ | 6.7 |
| L-ODE-ODE | $0.829 \pm 0.004$ | 22.0 |
| IP-Nets | $0.819 \pm 0.006$ | 1.3 |
| **mTAND-Enc** | $\mathbf{0.854 \pm 0.001}$ | **0.08** |
| **mTAND-Full** | $\mathbf{0.858 \pm 0.004}$ | **0.19** |

per training epoch in minutes. We note that the ODE-based models require substantially more run time than other methods due to the required use of an ODE solver (Chen et al., 2018; Rubanova et al., 2019). As we can see, the proposed full multi-time attention network (mTAND-Full) is over 85 times faster than ODE-RNN and over 100 times faster than L-ODE-ODE, the best-performing ODE-based models.

## 5. Discussion and Conclusions

In this paper, we have presented the multi-time attention (mTAN) module for learning from sparse and irregularly sampled data along with a VAE-based encoder-decoder model leveraging this module. Our results show that the resulting model performs as well or better than a range of baseline and state-of-the-art models on both the interpolation and classification tasks, while offering training times that are one to two orders of magnitude faster than previous state of the art methods.

While we have focused on a VAE-based encoder-decoder architecture, the proposed mTAN module can be used to provide an interface between sparse and irregularly sampled time series and many different types of deep neural network architectures including GAN-based models. Composing the mTAN module with convolutional networks instead of recurrent architectures may also provide further computational enhancements due to improved parallelism.

## Acknowledgements

Research reported in this paper was partially supported by the National Cancer Institute of the National Institutes of Health under award number 5U01CA229445-02.

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

# A. Appendix

## A.1. Dataset Description

The PhysioNet Challenge 2012 dataset (Silva et al., 2012) consists of multivariate time series data with 37 variables extracted from intensive care unit (ICU) records.[1] Each record contains sparse and irregularly spaced measurements from the first 48 hours after admission to ICU. We follow the procedures of Rubanova et al. (2019) and round the observation times to the nearest minute. This leads to 2880 possible measurement times per time series. The data set includes 4000 labeled instances and 4000 unlabeled instances. We use all 8000 instances for interpolation experiments and the 4000 labeled instances for classification experiments. We focus on predicting in-hospital mortality. 13.8% of examples are in the positive class.

## A.2. Experimental Protocols

We conduct interpolation experiments using the 8000 data cases in the PhysioNet data set. We randomly divide the data set into a training set containing 80% of the instances, and a test set containing the remaining 20% of instances. We use 20% of the training data for validation. During training, we condition on all observed time points. For all baseline models, we follow the same testing procedure used in Rubanova et al. (2019). The values at all time points in a test instance are conditioned on and each model is used to reconstruct them. We repeat each experiment five times using different random seeds to initialize the model parameters. We assess interpolation performance using mean squared error (MSE).

We use the labeled data to conduct classification experiment. We focus on whole time series classification. We randomly divide each data set into a training set containing 80% of the time series, and a test set containing the remaining 20% of instances. We use 20% of the training set for validation. We repeat each experiment five times using different random seeds to initialize the model parameters. Due to class imbalance in the PhysioNet data set, we assess classification performance using area under the ROC curve (the AUC score).

For both interpolation and classification tasks, we select hyper-parameters on the held-out validation set using grid search, and then apply the best trained model to the test set.

## A.3. Models

The model we focus on is the encoder-decoder architecture based on the discretized multi-time attention module (**mTAND-Full**). In the classification experiments, the hid-

den state at the last observed point is passed to a two-layer binary classification module for all models. For each data set, the structure of this classifier is the same for all models. For the proposed model, the sequence of latent states is first passed through a GRU and then the final hidden state is passed through the same classification module. For the classification task only, we consider an ablation of the full model that uses the proposed mTAND encoder, which consists of our mTAND module followed by a GRU to extract a final hidden state, which is then passed to the classification module (**mTAND-Enc**). We compare to several deep learning models that expand on recurrent networks to accommodate irregular sampling. We also compare to several encoder-decoder approaches. The full list of model variants is briefly described below. We use a Gated Recurrent Unit (GRU (Chung et al., 2014)) module as the recurrent network throughout.

- **RNN-Impute:** Missing observations replaced with weighted average of last observed measurement within that time series and global mean of the variable across the training examples (Che et al., 2018).

- **RNN-$\Delta_t$:** Input is concatenated with masking variable and time interval $\Delta_t$ indicating how long the particular variable is missing.

- **RNN-Decay:** RNN with exponential decay on hidden states (Mozer et al., 2017; Che et al., 2018).

- **GRU-D:** combining hidden state decay with input decay (Che et al., 2018).

- **IP-Nets:** Interpolation prediction networks, which use several semi-parametric RBF interpolation layers, followed by a GRU (Shukla & Marlin, 2019).

- **ODE-RNN:** Uses neural ODEs to model hidden state dynamics and an RNN to update the hidden state in presence of a new observation (Rubanova et al., 2019).

- **RNN-VAE:** A VAE-based model where the encoder and decoder are standard RNN models.

- **L-ODE-RNN:** Latent ODE where the encoder is an RNN and decoder is a neural ODE (Chen et al., 2018).

- **L-ODE-ODE:** Latent ODE where the encoder is an ODE-RNN and decoder is a neural ODE (Rubanova et al., 2019).

---

[1]https://physionet.org/content/challenge-2012/