# OpenReview forum: "Multi-Time Attention Networks for Irregularly Sampled Time Series"
_ICML.cc/2020/Workshop/Artemiss — ICML Artemiss 2020_

### Official Review · AnonReviewer1 · 2020-06-18
**Interesting method and results, I recommend accept.**

**Rating:** 7
**Confidence:** 3

**Review:**

The paper proposes a method for irregular sampled time series which involves a multi-time-attention mechanism that learns to interpolate missing time series data by attending to encodings at multiple reference points. This is embedded in a probabilistic setup with a modified VAE objective.

Strengths:
- The paper is clearly written and understandable.
- There is one extensive experiment with multiple baselines (interesting further canditates for the future: SeFT, BRITS, GP-VAE, MGP-RNN, MGP-TCN)
- Impressive runtime

Weaknesses:
- Only 1 dataset, whereas the quite imbalanced classification task is evaluated only with AUC (AUPRC would be more meaningful)

Further comments: I wonder to which degree this approach differs from a sparse GP interpolation with learnt query points, this could be experimentally but also theoretically interesting.

Minor comments:
-last paragraph Sec 1, and first paragraph Sec 2: re-represent --> represent

---

### Official Review · AnonReviewer2 · 2020-06-22
**Very interesting, relevant topic.**

**Rating:** 7
**Confidence:** 3

**Review:**

The workshop submission addresses multivariate, irregularly sampled time series. The authors propose a preprocessing step they call ‘Multi-Time Attention Module’, that represents the irregularly spaced time series in a fixed-dimensional representation.


Strengths:
- Extremely interesting topic
- Well-written
- Originality

Weaknesses:
- Experiment description could be better
- No sharing of code for the experiments


RELEVANCE:

The topic itself is extremely interesting for the workshop. Irregularly sampled time series are in some way also missing data problems. Often not all variables are recorded for one time stamp. The topic itself is very up-to-date.

SIGNIFICANCE OF THE PROBLEM:

These are quite common problems in real-world applications. Dealing with these problems can be quite complicated.

ORIGINALITY OF THE WORK:

There is novelty to the work. The paper itself is about really up-to-date research questions.

WRITING QUALITY:

Language and grammar of the submitted short paper are good.

TECHNICAL SOUNDNESS:

The short review time did not enable an in-depth review. The overall topic is relevant to the workshop and the authors make a legit attempt at contributing to the field. Yet, due to not performing an in-depth review I can’t fully discard that there might still be issues with the experiments.

REPLICABILITY:

In my opinion researchers in general should have the confidence to share their code used for conducting the experiments. Which was not the case here. Unfortunately, additionally some questions about the experiments remain open for me (despite the authors efforts in the appendix).

---

### Decision · Program_Chairs · 2020-07-02

**Decision:**

Accept

**Comment:**

We are very happy to inform you that your paper has been accepted for the Artemiss workshop. We will contact you soon to inform you about the details concerning the format of your presentation at the workshop, and the camera-ready version deadline. Please take into account the referee's comments to write the camera-ready version.